# GL2GPU: Accelerating WebGL Applications via Dynamic API Translation to WebGPU

Submission Id: 6

## Abstract

WebGL has long been the prevalent API for GPU-accelerated graphics in web browsers, boosting 2D/3D graphical web applications. Despite widespread adoption, WebGL's programming model hinders its rendering performance on modern GPU hardware. To this end, WebGPU has been proposed as the next-generation API of GPU-accelerated processing in web browsers, exhibiting higher performance than WebGL. However, considering the complex logic of WebGL applications and the still-evolving WebGPU specification, statically migrating existing WebGL applications to WebGPU from source code is labor-intensive. To address this issue, we propose GL2GPU, an intermediate layer that dynamically translates WebGL to WebGPU at JavaScript runtime to improve rendering performance. GL2GPU addresses the inconsistencies between the WebGL and WebGPU programming models by emulating WebGL rendering states and leverages performance optimization mechanisms introduced by WebGPU to reduce the overhead of dynamic translation. Evaluation of three representative WebGL benchmarks shows that GL2GPU significantly enhances end-to-end rendering performance while maintaining visual consistency, achieving an average frame time reduction of 45.05% across different devices and operating systems.

## CCS Concepts

• **Information systems** → **Browsers**; **Web applications**; • **Computing methodologies** → *Computer graphics*; • **Software and its engineering** → *Software notations and tools*.

## Keywords

Web applications; Graphics; WebGL; WebGPU; API translation

## 1 Introduction

Graphics rendering has become an increasingly popular and essential component of web applications, significantly enhancing user experiences in online multimedia domains such as gaming [54, 63], visual effects [17, 22, 32, 52, 56, 62], data visualization [13, 45, 55–57], and virtual reality [11, 19, 31, 34, 47].

To enable web developers to perform high-performance interactive 3D and 2D graphics rendering, the WebGL [29] API was introduced in 2009. WebGL employs an imperative API design that is straightforward and beginner-friendly, allowing developers to achieve hardware-accelerated graphics rendering with minimal code. As a result, a substantial base of applications has been developed using WebGL for over a decade. It is expected that new WebGL applications will continue to emerge.

Despite the success of WebGL, the programming model of WebGL struggles to meet the performance demands of modern graphics applications, which often involve heavy computational loads. To better leverage contemporary hardware capabilities, WebGPU was

proposed in 2017 [65, 68] as the next-generation web graphics API. WebGPU adopts a declarative programming model, which is more complex but offers significantly higher performance potential than WebGL. Recent work has shown that WebGPU substantially outperforms WebGL, particularly in applications with intensive rendering workloads [4, 5, 15, 23]. Therefore, migrating existing WebGL applications to WebGPU API can enhance rendering performance. Recognizing the potential of WebGPU, developers have started to rewrite WebGL applications using the WebGPU API [24, 66].

However, rewriting WebGL applications into WebGPU is labor-intensive, time-consuming, and error-prone. The differences in the programming models of these two APIs make trivial API rewriting difficult [42]. Furthermore, some rendering frameworks fail to utilize WebGPU's features during rewriting, leading to decreased performance [10]. This prevents existing WebGL applications from quickly and easily benefiting from the performance enhancements offered by WebGPU.

In this paper, we introduce GL2GPU, a dynamic WebGL-to-WebGPU translator in the JavaScript runtime. GL2GPU requires no modifications to the browser or the original logic of WebGL applications, enabling legacy and newly developed WebGL applications to benefit from WebGPU's performance improvements in an easy way. The basic idea of GL2GPU is to track imperative rendering state changes using JavaScript prototype patching, increase the reusability of rendering resources, and leverage WebGPU features to reduce runtime overhead. Specifically, GL2GPU addresses two significant challenges in the dynamic translation process: the inconsistency between the WebGL and WebGPU programming models (C1) and the high translation overhead within the JavaScript runtime (C2). We propose the following insights to tackle these challenges:

• (C1) *Imperative changes in WebGL can be tracked by JavaScript prototype patching.* GL2GPU introduces a novel state-tracking mechanism to conduct the translation. We build the model of WebGL and emulate the global render state in JavaScript runtime. GL2GPU tracks the imperative rendering context configuration in WebGL by injecting code into the prototype of the WebGL API. Additionally, with the help of this rendering state, GL2GPU analyzes the WebGL shaders and merges shared variables to generate the corresponding WebGPU shaders accurately.

• (C2) *Caching previously encountered WebGL states reduces the overhead of traversing.* GL2GPU proposes a novel representation of WebGL rendering state transition. GL2GPU introduces a state transition management algorithm based on a cache mechanism. This caching design significantly boosts the performance of tracking state changes.

• (C2) *Utilizing bundles can reduce the overhead of recording rendering commands in the JavaScript runtime.* GL2GPU introduces a bundle management algorithm to improve the reusability of recorded WebGPU operations, reducing the interpretation overhead of the JavaScript source code.

We utilize the widely used WebGL benchmarks for our evaluations, using the native WebGL implementation within the browser as our baseline. Our consistency assessment, conducted through pixel-by-pixel comparison, confirms that GL2GPU maintains high visual consistency. Our performance assessment demonstrates significant improvements in rendering times, with the average frame time reduced by 45.05% across various devices compared to the baseline. The minimum frame time reduction is 3.3%, and the maximum frame time reduction is 87.7%. The ablation study also validates the effectiveness of the proposed optimization mechanisms. In summary, the main contributions of our work are as follows:

- We propose GL2GPU, an intermediate layer to dynamically translate WebGL invocations to WebGPU at JavaScript runtime by prototype patching. This seamless translation leverages the advanced capabilities of WebGPU, enabling existing WebGL applications to benefit from improved performance without extensive modifications.
- We design novel optimization mechanisms to reduce the runtime overhead during dynamic API translation. These optimizations leverage the unique features of WebGPU to boost the translation process, enhancing overall efficiency.
- We evaluate GL2GPU's consistency and performance improvements on representative benchmarks over various devices. Our further ablation study validates the effectiveness of key designs.

We provide background on graphics rendering (§ 2), present the design (§ 3) , conduct an evaluation (§ 4), discuss related work (§ 5), and conclude our work (§ 6). To foster further research on this topic, we release the source code of GL2GPU at https://anonymous.4open.science/r/gl2gpu-E381/.

## 2 Rendering Process in Web Apps

Rendering in a web application involves sequentially processing each object. As shown in Figure 1, the object rendering workflows of WebGL and WebGPU can be briefly summarized as follows: shaders take data as input and produce the final image output. A rendering context configures this process. The GPU generates many threads to enhance parallelism while rendering an object, each executing the same shader and sharing the same rendering context. These threads compute the color of each pixel in the output image concurrently. We elaborate more on the details of WebGL and WebGPU rendering processes in Appendix A.

- *Shaders* encapsulate the core computational logic for the rendering process. Executed directly by the graphics processing unit (GPU), shaders perform complex rendering calculations efficiently. In WebGL and WebGPU, the shading process is divided into two distinct parts: *vertex shading*, which maps input vertices to their corresponding coordinates, and *fragment shading*, which performs interpolation between pixels.
- The *inputs* of a shader contains two parts: shared variables and local variables. The *shared variables*, shared among all threads, include uniform buffers and textures. Uniform buffers store a handful of numbers, such as lighting direction and camera matrices. Textures contain texture images. *Local variables* includes vertex buffers and index buffers and is unique to each thread. Vertex buffers contain information about each vertex, such as coordinates and colors. Index buffers detail the drawing order of vertices.

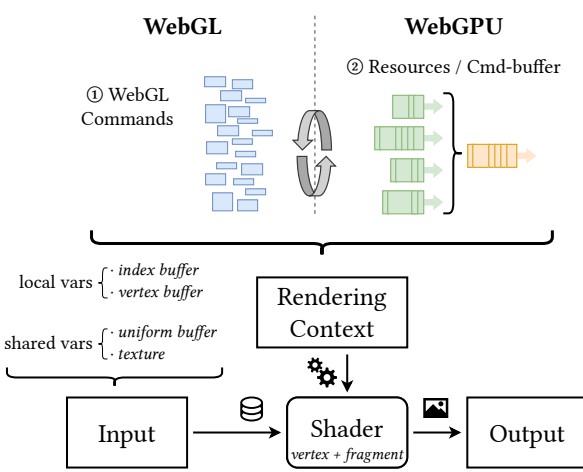

**Figure 1: Comparison between WebGL and WebGPU.**

- The *rendering context* configures details related to rendering. These include data bindings, specifying from which buffers the shader's inputs are retrieved; fragment shader configurations, such as the enabling of face culling and depth testing; the layout of the vertex buffer; and more.

The differences between WebGL and WebGPU programming models lie in how they configure the rendering context. As shown in Figure 1, the rendering context of WebGL is configured through the imperative setting of a global state machine (①). In contrast, the rendering context of WebGPU is set by declaratively setting rendering resources in a command buffer and submitting the encoded command to GPU (②). Meanwhile, WebGL utilizes **GLSL** (OpenGL Shading Language) for its shaders, while WebGPU uses **WGSL** (WebGPU Shading Language). In GLSL, shared variables between vertex and fragment shaders are aligned based on their variable names. In contrast, WGSL requires developers to assign a unique, incrementing location ID number to each shared variable, starting from 0. This inconsistency in programming models makes API translation based on static analysis a hard problem [42].

Another distinction between WebGL and WebGPU is their granularity of resource management. WebGL's resource management is much coarser compared to that of WebGPU. Furthermore, WebGPU introduces a unique feature tailored for the JavaScript environment: the rendering bundle. Due to the dynamic nature of JavaScript, where function calls can be time-consuming, WebGPU's GPURenderBundle represents a partially recorded rendering configuration that can reduce the time spent on JavaScript function calls. Therefore, effectively harnessing the reusability of WebGPU resources is crucial for improving rendering performance.

## 3 Design

This section introduces the design of GL2GPU, including an overview of its workflow and module details.

### 3.1 Overview

The workflow of GL2GPU is illustrated in Figure 2. GL2GPU captures WebGL invocations from the web application and translates them into WebGPU commands. Achieving an effective translation

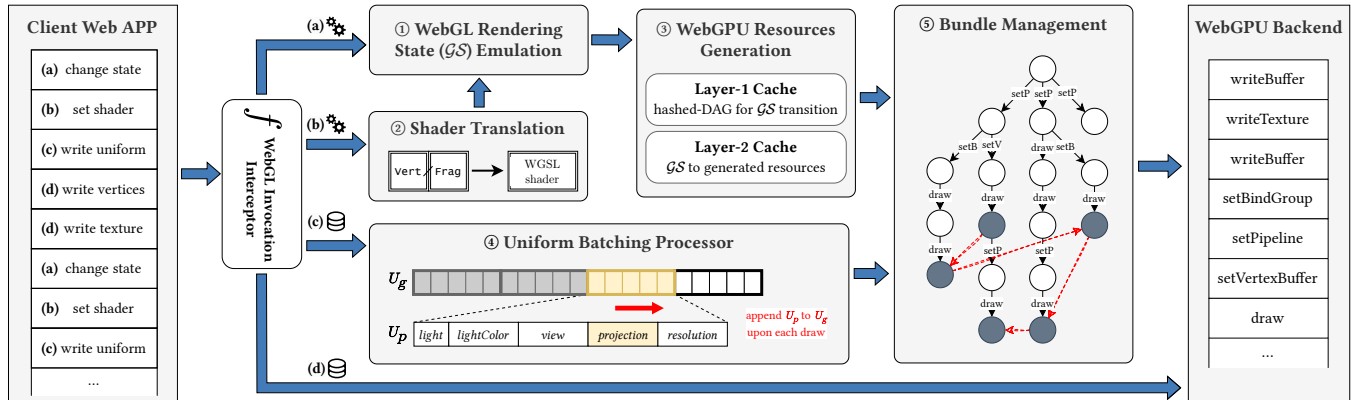

**Figure 2: Overall workflow of GL2GPU.**

from WebGL to WebGPU within a JavaScript runtime relies on two key aspects: maintaining rendering **consistency** and improving translation **performance**. The basic modules of GL2GPU, labeled ① through ⑤, address these aspects. Among them, ① and ② are the modules to ensure the translation consistency. ③, ④ and ⑤ are three modules to improve the translation performance. We provide detailed descriptions of these five modules in the remainder of this section.

GL2GPU classifies the WebGL invocations into four main categories and deals with them differently. These categories include (a) drawing and state changes, (b) shader compilation, (c) uniform updates, and (d) vertices/texture data uploads. Precisely, GL2GPU emulates a WebGL rendering state (①), represented as $\mathcal{GS}$. For type (a) invocations, specific sections within $\mathcal{GS}$ are modified. For type (b), GL2GPU utilizes a mechanism to merge shared variables from GLSL shaders, resulting in the final translated WGSL shader (②). We design a fast WebGPU resource generator, where the compiled WGSL code and associated WebGPU resources are cached (③). For type (c), updates to uniforms are batched to reduce frequent GPU memory access (④). For type (d), modifications to vertex and texture buffers are directly uploaded to the GPU. WebGPU resource and uniform operations are packed using GL2GPU's render bundle mechanism (⑤). During drawing operations, GL2GPU retrieves rendering instructions from the render bundle, encodes them into GPU commands, and submits GPU commands to the WebGPU backend.

### 3.2 Emulating WebGL Rendering Context

WebGL developers set rendering context imperatively by modifying the GL rendering context [61]. Therefore, GL2GPU needs to track the rendering context changes and translate the context into the WebGPU descriptors. GL2GPU maintains an emulated WebGL global rendering context, denoted as $\mathcal{GS}$. This rendering context contains the basic components of WebGL, like the textures, vertex arrays, uniform buffers, programs, and frame buffers [29].

Due to JavaScript's dynamic nature, we use prototype patching to capture native WebGL API calls. Upon capturing state change invocations from WebGL (Figure 2, a), GL2GPU reflects these modifications within $\mathcal{GS}$ (Figure 2, ①). When drawing objects, GL2GPU generates the WebGPU descriptors based on $\mathcal{GS}$. Although many

intermediate rendering contexts can be encountered through executing a WebGL application, many of these rendering contexts may never be used. In fact, generating the correct WebGPU descriptors at the moment of drawing is essential. For details on the implementation, please refer to our open-source code.

### 3.3 Translating Shaders

WebGL's GLSL shaders must be translated to WebGPU's WGSL shaders before GL2GPU can create the correct WebGPU resource descriptor. The translation from GLSL shaders to WGSL shaders encompasses two main aspects: syntax translation and shared variables' alignment. On the one hand, considerable effort has been devoted to addressing syntax translation. For instance, rendering frameworks such as Playcanvas [53] and Babylon [6] utilize a WebAssembly module to implement a GLSL lexer within their framework. Considering that shaders in real-world applications are limited and often generated by frameworks through concatenation, we can pre-translate GLSL shaders to WGSL. This allows for a direct lookup of the pre-translated WGSL during dynamic translation.

On the other hand, aligning shared variables is not straightforward. In GLSL, shared variables are aligned based on their names, while WGSL requires developers to assign a unique number to each shared variable. The WGSL compiler uses these location numbers to align variables from vertex and fragment shaders. Given that different combinations of vertex and fragment shaders result in varying sets of shared variables, the assignment of WGSL location numbers also varies.

As shown in Figure 3, GL2GPU employs a merging mechanism to address this challenge. GL2GPU analyzes the GLSL shaders when the web application sets the shader source, generating shared variables within the GLSL. Meanwhile, the main logic of the GLSL shader is converted into a partially translated WGSL, which does not include any shared variables. When the web application links the vertex and fragment shaders together, GL2GPU combines the records from both shaders. This mixed record is concatenated with the partially translated WGSL code to generate a complete WGSL shader. Finally, this complete WGSL shader will be attached to the WebGPU pipeline descriptor when translating the WebGL render state.

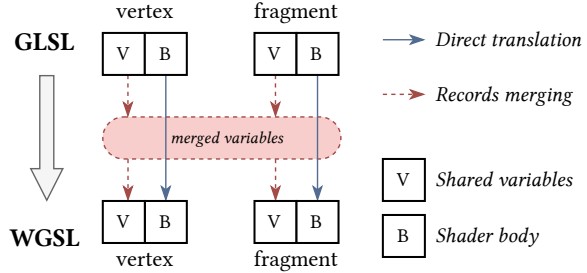

**Figure 3: Workflow of shared variable merging in GL2GPU.**

## 3.4 Generating WebGPU Resources

Generally, it is necessary to traverse the emulated WebGL rendering states when generating declarative WebGPU descriptors. However, this rendering state is quite complex, and such traversal is costly in JavaScript runtime. To this end, GL2GPU employs a two-layer caching strategy (Figure 2, ③) to retrieve WebGPU resources that correspond to the WebGL rendering state $\mathcal{GS}$ efficiently.

The first caching layer maintains a hashed-directed acyclic graph (DAG) to track the rendering state transitions. These transitions are WebGL invocations that change the rendering state. We present the pseudo-code in Algorithm 1. Specifically, in this DAG, each node represents a $\mathcal{GS}$ and its corresponding cached WebGPU resources; each edge represents a WebGL state-changing function $f$. When GL2GPU captures a WebGL state-changing invocation $f$ (line 5), it searches within this DAG for an edge that originates from the current rendering state ($\mathcal{GS}$) and is associated with function $f$. If such an edge $e = \langle v_{\mathcal{GS}}, v_{\mathcal{GS}'}, f \rangle$ is located (line 10), GL2GPU directly transitions to the target rendering state $\mathcal{GS}'$. If no such edge exists, a new vertex $v_{\mathcal{GS}'}$ is created, and the edge $e = \langle v_{\mathcal{GS}}, v_{\mathcal{GS}'}, f \rangle$ is added to the DAG (line 13-16). The introduction of this first caching layer mitigates the overhead associated with WebGL state transitions.

---

**Algorithm 1** Traversal on the DAG in the layer-1 cache.

---

1: **Variables:**
2: $\mathcal{GS}$: Current WebGL global rendering state
3: $G$: The hashed-DAG in the layer-1 cache.
4: *cur*: The DAG node corresponding to current $\mathcal{GS}$
5: **upon** WebGL state-changing invocation $f$ captured, **call** *recordDAG*($f$)
6: **External Function:**
7: transition($\mathcal{GS}$, $f$): returns the new rendering state after applying WebGL invocation $f$ to rendering state $\mathcal{GS}$
8: **function** RECORDDAG($f$)
9:     $\mathcal{GS}' \leftarrow$ transition($\mathcal{GS}$, $f$)
10:     **if** edge $\langle cur, m, f \rangle \in G$ **then**
11:         $cur \leftarrow m$
12:     **else**
13:         create a new DAG vertex $n$
14:         $n.\mathcal{GS} \leftarrow \mathcal{GS}'$
15:         add edge $\langle cur, n, f \rangle$ to $G$
16:         $cur \leftarrow n$
17:     **end if**
18:     $\mathcal{GS} \leftarrow \mathcal{GS}'$
19: **end function**

---

The second caching layer maintains a mapping from a $\mathcal{GS}$ to previously generated WebGPU resources. If a WebGL state change invocation misses in the first caching layer, we compute the hash of the $\mathcal{GS}$ and search for the corresponding WebGPU resources in this second caching layer. It is important to note that computing the hash of a $\mathcal{GS}$, especially in JavaScript, is highly time-consuming. Therefore, the introduction of the second caching layer reduces the overhead associated with regenerating WebGPU descriptors and WebGPU resources for the rendering states that have previously occurred.

## 3.5 Batching Uniform Updates

Each object's uniform buffer must be transferred from the CPU memory to the GPU memory while rendering a scene. Although individual uniform buffers are small, the frequent need for communication leads to substantial overhead. To mitigate this, GL2GPU leverages the "dynamic uniform offset" feature of WebGPU that enables the merging of uniform buffers for different object drawings.

WebGPU allows developers to specify the offset for the current draw's uniform within the uniform buffer. By leveraging this feature, we can batch the uniforms for multiple draw operations into a single uniform buffer. GL2GPU maintains a *global* uniform buffer on the CPU side to store the uniform buffers of distinct draw calls (Figure 2, ④). For each WebGL program $p$, GL2GPU maintains a uniform buffer on the CPU side. For convenience, we denoted the global uniform buffer as $U_g$ and the uniform buffer of a WebGL program $p$ as $U_p$. GL2GPU monitors updates to each WebGL program $p$'s uniform buffer (Figure 2, d) and applys these changes to the corresponding $U_p$. When a WebGL draw call is invoked (Figure 2, a), GL2GPU appends the active WebGL program's $U_p$ to the end of the global uniform buffer $U_g$. The active WebGL program $p$ is identified by tracking the most recent gl.useProgram($p$) call. Once $U_g$ is filled, GL2GPU uploads it to the GPU and clears it. Both the updates to $U_p$ and the appending to $U_g$ are executed on the CPU side without transferring data from the CPU to the GPU.

## 3.6 Managing Render Bundles

After the generation of WebGPU resources, GL2GPU generates WebGPU operations and packs them in a WebGPU bundle (Figure 2, ⑤). The key to performance enhancement brought by WebGPU bundles lies in their reusability. To enhance the reusability of WebGPU bundles, GL2GPU employs a Trie structure [41] to organize these generated bundles. Rendering an object involves a series of consecutive operations starting from scratch. The Trie facilitates efficient search and retrieval capabilities for rendering different objects by treating the sequence of rendering operations as strings.

As shown in Figure 2, the Trie structure is designed such that each node represents a render bundle. Each edge represents a WebGPU operation (e.g., **draw**, **setP**ipeline, **setV**ertexBuffer and **setB**indGroup) that can be packed into a render bundle. The root node denoted as $r$, represents an empty render bundle. The path from the root to any given node delineates a sequence of WebGPU operations, with the order of these operations corresponding to the order of the edges along the path (starting from the root node). These operations could be recorded in a bundle, which could be attached to the node for future reuse. The red arrows in Figure 2 (⑤)

demonstrate the sequence for five objects in a scene. The first and second renderings share the same setPipeline() operation. There are no shared preprocessing operations between the second and third renderings, and so forth. This structure allows for efficient reuse of rendering operations sequence across objects.

---

**Algorithm 2** Management of WebGPU operations and bundles using a Trie structure.

---

1: **Variables:**
2: $T$: Trie used for managing WebGPU operations and bundles.
3: $r$: Root node of $T$, representing an empty render bundle.
4: $p$: Pointer to the "current node" in $T$, initially set to $r$.
5: $*p$: The "current node". We have $\&(*p) = p$.
6: **upon** new WebGPU operation $op$ is generated, call $recordTrie(op)$
7: **upon** uniforms are flushed, call $execute()$

8: **function** RECORDTRIE(op)
9:   **if** no outgoing edge from $p$ labeled $op$ exists **then**
10:     $n \leftarrow$ Create a new Trie node
11:     Insert edge $\langle p, n, op \rangle$ into $T$
12:     $p \leftarrow \&n$
13:   **else**
14:     $p \leftarrow \&m$ where $\langle p, m, op \rangle \in T$        ▷ Move $p$ to the node $m$ connected by $op$
15:   **end if**
16: **end function**

17: **function** EXECUTE
18:   **if** WebGPU bundle $b$ is not attached to node $*p$ **then**
19:     Generate WebGPU bundle $b$ recording all operations on edges along the path from $r$ to $*p$
20:     Attach $b$ to node $*p$
21:   **end if**
22:   Execute WebGPU bundle $b$ attached to node $*p$
23:   Update $p$ to point to root node $r$
24: **end function**

---

Algorithm 2 describes the node management of this Trie. GL2GPU maintains a pointer $p$, where the node pointed to by $p$ (represented as $p*$) is referred to as the "current node". Initially, the Trie consists solely of the root node $r$, with $p$ pointing to $r$. Upon generating a new WebGPU operation $op$ (line 6), GL2GPU first checks whether $*p$ has an outgoing edge labeled $op$. If such an edge does not exist, GL2GPU creates a new Trie node $n$ and inserts the edge $\langle *p, n, op \rangle$ into the Trie (line 10). Then, an edge $\langle *p, m, op \rangle$ must exist. We update $p$ to point to node $m$ (line 12 and 14).

GL2GPU generates at least four WebGPU operations at the end of an object drawing invocation (Figure 2, a). These operations include setBindGroup, setPipeline, setVertexBuffer, and draw (or drawIndexed). However, the previously mentioned uniform batching mechanism delays the update of uniforms, meaning the uniforms in GPU memory still need to be updated. Therefore, the submission of WebGPU operations to the GPU is also postponed to ensure the accuracy of the rendered objects. As a result, GL2GPU packs WebGPU operations from multiple drawing invocations into a single WebGPU bundle. GL2GPU executes this WebGPU bundle once the global uniform buffer is flushed to GPU. Upon the global uniform is flushed (line 7), GL2GPU executes all WebGPU operations along the path from the root node $r$ to $*p$ (line 19). If

no WebGPU bundle is associated with $*p$, GL2GPU creates a new WebGPU bundle $b$, attaches this bundle to the corresponding Trie node (line 20), and executes (line 22) this bundle. After executing the render bundle, GL2GPU set $p$ to point back to the root node $r$ (line 23) for the next drawing.

## 4 Evaluation

We evaluate our translator with the following questions:

- **RQ1: Consistency.** Does GL2GPU maintain consistency in rendering results?
- **RQ2: Performance.** How is the scalability of GL2GPU's performance improvement? How does this improvement vary across different devices?
- **RQ3: Ablation Study.** Are the proposed optimization mechanisms effective?

### 4.1 Experiment Setup

**Implementation.** We implement GL2GPU as a Node.js package, comprising approximately 5,200 lines of TypeScript code. We employ Webpack [7] to bundle this into a standalone JavaScript file, facilitating direct integration of GL2GPU into applications. For example, developers can embed it using the <script> tag in HTML or import this JavaScript module into a browser extension.

**Benchmarks.** Following existing work on WebGL graphics rendering, we evaluate the consistency and performance of GL2GPU on three well-known benchmarks in the web ecosystem. These benchmarks cover various rendering techniques, including 2D and 3D rendering, texturing, lighting, and shadows. The details of these benchmarks are as follows:

- *MotionMark* [3]. Developed by the WebKit team, this benchmark measures a browser's capacity to handle complex animations. We use the WebGL performance test included in this benchmark to assess the performance of the rendering performance capability. This WebGL performance test renders many triangles with gradient colors but without binding textures. It primarily evaluates the performance of increasing the number of vertices.
- *JSGameBench* [20]. Introduced by Meta in 2011 [1], this benchmark evaluates web gaming performance by simulating heavy yet adjustable workloads. It renders a large number of textured sprites with configurable sprite counts.
- *Aquarium* [2]. This WebGL benchmark stems from a real-world application, which renders an aquarium with complex 3D models, textures, and lighting effects. Previous research utilized this benchmark to evaluate the performance of WebGL/OpenGL ES implementations [59] or conduct object segmentation [64]. It is also frequently discussed in the community, with over 200 issues in the Chromium issue tracker related to a specific version failing to run this app. The success of this application has also drawn industry attention, as seen in Intel's native implementation of this web app to measure the performance of native graphics rendering [36].

We added a <src> element to each benchmark's HTML source code to conduct our evaluation.

**Devices.** We selected a range of commonly used devices from the market to ensure a comprehensive analysis. For Apple devices, our selection includes a MacBook Pro with an M1 chip and another

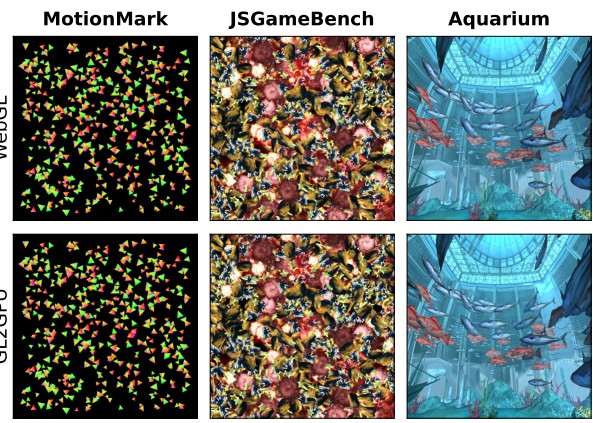

**Figure 4: Comparison of output before and after translation.**

with an M3 chip. For PC devices, we equipped our workstation with a high-performance AMD 6900XT GPU and another with an NVIDIA RTX 3070 GPU. Additionally, we are using a mini-PC with an integrated AMD 7840HS GPU, a laptop featuring a discrete NVIDIA GTX 1650 GPU, and another laptop with an integrated Intel i5-8265U GPU. For mobile devices, our study includes a Redmi K60 powered by a Qualcomm Snapdragon 8 gen 2 SoC, an Oppo Find X3 with a Snapdragon 870 and a Redmi Note 11T Pro with a Dimensity 8100.

**Browsers.** We conduct experiments in both Firefox and Chrome; however, we encounter issues due to the incomplete implementation of the basic WebGPU functionality within Firefox [50]. We have reported the issue to Firefox developers, who have confirmed and are currently addressing it. Consequently, we select Chrome version 114.0.5735, the earliest version supporting WebGPU, as our testing browser.

**Baseline.** To the best of our knowledge, GL2GPU is the first to focus on improving WebGL application's rendering performance by translating to WebGPU. Therefore, we utilize the native WebGL implementation provided by the browser as the baseline.

## 4.2 Consistency

We follow the established pixel-by-pixel comparison method in the field of computer graphics [8, 16, 37, 48] for consistency assessment to evaluate whether GL2GPU faithfully finishes the original rendering tasks. We utilize the Mean Squared Error (MSE) as the error metric. Given two images of identical dimensions, where $I_x$ represents the original image and $I_y$ is the generated image, both with dimensions $M \times N \times C$ (width $M$, height $N$, and color channels $C$), the MSE is calculated as shown in Equation (1):

$$\text{MSE}\left(I_x, I_y\right) = \frac{1}{M \cdot N \cdot C} \sum_{i=0}^{M-1} \sum_{j=0}^{N-1} \sum_{k=0}^{C-1} \left(I_x(i,j,k) - I_y(i,j,k)\right)^2 \quad (1)$$

In Equation (1), $I_x(i, j, k)$ and $I_y(i, j, k)$ denote the intensity of the $k^{th}$ color channel at pixel position $(i, j)$ in the original and generated images, respectively. Each color channel is a floating-point number, scaled between 0 and 1. We develop a tool for this

evaluation that waits until the page has fully loaded and rendering initialization is complete before saving $N$ ($N = 100$) consecutive frames of canvas contents. Since the traces from WebGL and GL2GPU belong to different runs, we match the frames captured from both to compute the MSE.

We modified the benchmarks to minimize inconsistencies across multiple runs. The modifications are as follows: (1) We disabled antialiasing and set the HTML canvas element size to 1024 pixels for both height and width. (2) We replace the default random number generator `Math.random()` with a pseudo-random number generator, which ensures the controllable generation of random numbers. (3) We replace the `performance.now()` with a linear function. This JavaScript function originally generates a high-resolution timestamp in milliseconds, which fails to deliver consistent return values in different runs.

Figure 4 illustrates examples of the canvas content of frames before and after translation for the three benchmarks, showing the matched frames. The comparison reveals minimal differences between the WebGL-rendered images and those generated by GL2GPU. Specifically, the average MSE across 100 consecutive frames is $1.50 \times 10^{-3}$ for the *MotionMark* benchmark, $1.87 \times 10^{-3}$ for the *JSGameBench* benchmark, and $5.37 \times 10^{-3}$ for the *Aquarium*. A similar pixel-by-pixel comparison of images rendered on a Mac-Book M1 and a PC with an AMD 6900XT using the baseline WebGL for the *Aquarium* benchmark showed an MSE of $5.89 \times 10^{-3}$. The MSE error of GL2GPU is smaller than the difference observed between two hardware platforms, validating GL2GPU's effectiveness in maintaining visual consistency.

## 4.3 Performance

*4.3.1 Scalability.* Following previous research [10, 14, 15, 27], we measure the time spent on rendering a single frame (denoted as frame time, FT) to evaluate the performance improvement delivered by GL2GPU. A lower FT indicates better performance. We adjust the complexity of the rendering scene by changing the number of rendered objects. We compare the end-to-end rendering performance of the original WebGL backend, denoted as $t_{\text{webgl}}$, with the performance achieved by our method, denoted as $t_{\text{gl2gpu}}$.

Figure 5 presents the evaluation results, which contains three subfigures, each illustrating the FT results of our evaluation across varying object counts within the *MotionMark*, *JSGameBench*, and *Aquarium*. The $x$-axis denotes the number of objects rendered: triangle number in *MotionMark*, fish number in the *Aquarium*, sprite number in *JSGameBench*. The $y$-axis of each subfigure denotes the averaged FT in milliseconds. The performance trends with varying object counts are similar across different devices. Therefore, we only present the results drawn from the MacBook Pro M1 due to space constraints.

The results indicate that GL2GPU demonstrates a pronounced improvement in end-to-end rendering performance compared to native WebGL as the rendering backend. This improvement becomes evident, especially as the rendering scene becomes complicated. Moreover, we also find that the $t_{\text{webgl}}$ tends to increase linearly as the number of rendered objects increases. However, for $t_{\text{gl2gpu}}$, we observe that the slope of the FT is not a constant in *MotionMark*. For instance, in Figure 5(a), the slope decreases between $x = 0.7$ to

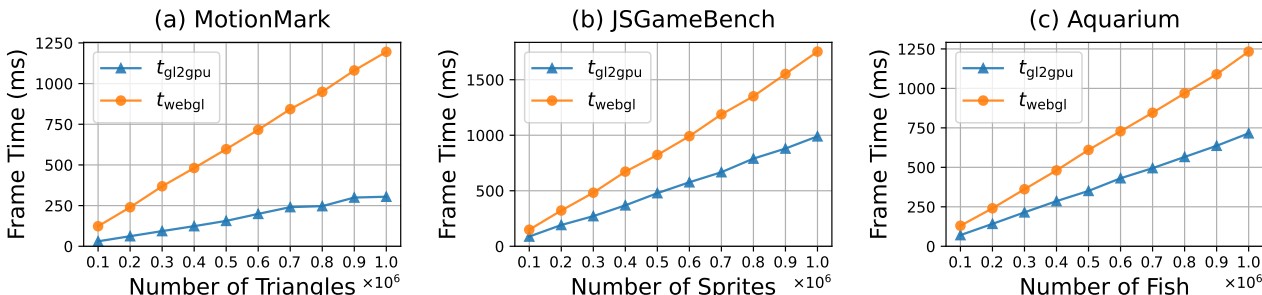

Figure 5: Frame times across different benchmarks with varying object numbers (lower is better).

Table 1: Frame times (FT, in milliseconds) and reduction in FT across multiple devices. The columns labeled $t_{webgl}$ and $t_{gl2gpu}$ represent the FT in milliseconds for the WebGL baseline and GL2GPU, respectively. Columns marked with ↓ indicate the reduction in FT.

| OS | Device | MotionMark | | | JSGameBench | | | Aquarium | | |
|---|---|---|---|---|---|---|---|---|---|---|
| | | $t_{webgl}$ | $t_{gl2gpu}$ | ↓ | $t_{webgl}$ | $t_{gl2gpu}$ | ↓ | $t_{webgl}$ | $t_{gl2gpu}$ | ↓ |
| MacOS | MacBook Pro M3 | 482.4 | 125.0 | **74.1**% | 647.7 | 351.8 | **45.7**% | 497.5 | 280.2 | **43.7**% |
| MacOS | MacBook Pro M1 | 596.9 | 156.0 | **73.9**% | 822.7 | 478.3 | **41.9**% | 610.3 | 350.6 | **42.6**% |
| Windows | PC with AMD 6900XT | 280.8 | 121.6 | **56.7**% | 615.7 | 410.0 | **33.4**% | 268.4 | 172.3 | **35.8**% |
| Windows | PC with Nvidia RTX3070 | 320.4 | 125.8 | **60.7**% | 646.6 | 522.1 | **19.3**% | 249.1 | 184.1 | **26.1**% |
| Windows | PC with AMD 7840HS | 420.4 | 166.7 | **60.3**% | 956.2 | 885.0 | **7.4**% | 239.4 | 231.5 | **3.3**% |
| Windows | PC with Nvidia GTX1650 | 970.1 | 395.5 | **59.2**% | 1,912.4 | 1,798.3 | **6.0**% | 696.9 | 491.9 | **29.4**% |
| Windows | PC with Intel 8265U | 1,162.9 | 362.6 | **68.8**% | 3,069.5 | 2,383.3 | **22.4**% | 892.3 | 553.4 | **38.0**% |
| Android | Mobile Redmi K60 | 2,406.2 | 295.3 | **87.7**% | 2,058.4 | 1,048.4 | **49.1**% | 1,756.8 | 763.9 | **56.5**% |
| Android | Mobile Oppo Find X3 | 2,601.4 | 469.3 | **82.0**% | 3,366.9 | 1,298.9 | **61.4**% | 2,582.5 | 935.9 | **63.8**% |
| Android | Mobile Redmi Note 11T Pro | 1,412.0 | 440.3 | **68.8**% | 2,025.2 | 1,478.6 | **27.0**% | 1,174.1 | 1,098.7 | **6.4**% |

0.8 and 0.9 to 1.0. This phenomenon is because as scene complexity increases, GL2GPU can reuse existing WebGPU resources, which may prevent the rendering time from increasing linearly with scene complexity.

We present the detailed FT on this MacBook in the second row of Table 1, where the device name is "MacBook Pro M1". The performance gains achieved with GL2GPU are significant across all tested scenarios. At the highest tested object count of 500,000, the average FT for the *MotionMark* is reduced from 596.9 ms to 156.0 ms, equating to an FT reduction of 73.9%. Similar improvements are also observed in the *JSGameBench* and *Aquarium*, where GL2GPU reduced the average FT to 478.3 ms and 350.6 ms, respectively, thereby achieving FT reductions of 41.9% and 42.6% compared to the baseline. The results indicate that GL2GPU exhibits the most substantial improvement in the *MotionMark*, followed by *Aquarium*, and then *JSGameBench*.

The differences in the improvement are attributed to the differing complexities of the WebGL global state across benchmarks. *MotionMark* does not involve texture processing, and its global state transitions are infrequent, making CPU-GPU communication a performance bottleneck. GL2GPU enhances performance significantly by reusing resources, thus reducing the overhead of CPU-GPU communication. Both *JSGameBench* and *Aquarium* involve extensive texture processing. However, the scene's complexity in *Aquarium* necessitates complex WebGL global state configurations, leading to

frequent context switches that increase the CPU time when executing the WebGL baseline. GL2GPU mitigates this overhead through context caching and render bundle packaging. In contrast, the scene in *JSGameBench* does not require complex context switches, hence the limited performance improvement with GL2GPU.

*4.3.2 Adaption on Heterogeneous Devices.* We evaluated the performance improvement of GL2GPU on heterogeneous devices. Table 1 demonstrates the FT on three benchmarks when rendering 500,000 objects across our experiment devices, with each benchmark comprising three columns of data: the baseline FT for WebGL ($t_{webgl}$), the FT after translation via GL2GPU ($t_{gl2gpu}$), and the percentage of FT reduction achieved by GL2GPU (↓, calculated by $\frac{t_{webgl} - t_{gl2gpu}}{t_{webgl}} \times 100\%$). The evaluation results reveal that equipped with GL2GPU, all the devices show a markable improvement in rendering FT across different benchmarks. For instance, the MacBook M3 exhibited performance gains in all three benchmarks, most notably in the *JSGameBench* benchmark with a 46.82% improvement. Additionally, with the same operating system, an increase in hardware specs reduces the FT. The results further highlight the variability of GL2GPU's impact on reducing FT across different operating systems and devices.

The experimental results demonstrate that the average frame time was reduced by 45.05%. The frame time reduction varied across different operating systems. For instance, on the Windows platform,

**Table 2: Frame times (in milliseconds) across different GL2GPU configurations.**

| Benchmark | GL2GPU | No-C | No-L1C | No-GU | No-B |
|---|---|---|---|---|---|
| MotionMark | 156.0 | 9,060.2 | 1,131.1 | $> 10^5$ | 221.0 |
| JSGameBench | 478.3 | $> 10^5$ | 1,810.1 | $> 10^5$ | 571.8 |
| Aquarium | 350.6 | $> 10^5$ | 3,143.3 | $> 10^5$ | 354.6 |

a PC with an Nvidia GTX1650 GPU showed a 29.4% improvement in the *Aquarium* benchmark. In contrast, a laptop with an Intel 8265U integrated GPU showed a 38.0% improvement. On Redmi K60, GL2GPU achieved an average performance improvement of 64.43%, in which the FT of *MotionMark* decreased from 2,406.2 ms to 295.3 ms, representing an 87.7% improvement. This variation is primarily due to differences in the implementation of native graphics drivers (like OpenGL, Vulkan, and Direct3D) across different operating systems.

## 4.4 Ablation Study

We conduct an ablation study to validate the performance optimization mechanisms proposed in the design of GL2GPU. Table 2 presents the FT evaluation results for GL2GPU on the MacBook M1 device under five different configurations. Specifically, the column labeled "GL2GPU" contains the FT of GL2GPU with all optimization mechanisms. Column "No-C" represents FT with the cache mechanism removed; "No-L1C" denotes FT without the Layer-1 Cache; "No-GU" indicates FT without the uniform batching mechanism; "No-B" shows FT without the bundle management.

For *MotionMark*, GL2GPU with all optimizations enabled records an FT of 156.0 ms. Removing the bundle design results in an FT of 221.0 ms, while removing the cache design leads to an FT of 9,060.2 ms, and removing the first layer of cache results in an FT of 1,131.1 ms. Similar trends are observed in *JSGameBench* and *Aquarium*. Our findings indicate that removing the uniform optimization results in execution times exceeding 10 seconds across all three benchmarks. Similarly, removing the entire cache optimization leads to FT greater than 10 seconds in both *JSGameBench* and *Aquarium*. Only when removing the first caching layer or the bundle optimization does the FT drop below 10 seconds. Among these, the impact of removing the bundle optimization is significantly less detrimental than that of removing the first caching layer. However, regardless of the optimization mechanism removed, the final performance is worse than that of GL2GPU with all optimizations included. The results of our ablation study validate the effectiveness of the key design of GL2GPU.

## 5 Related Work

**WebGPU.** In recent years, there has been a surge in research focusing on WebGPU, aiming for shader testing, security enhancements, and rendering performance improvements. Levine et al. introduced a technique for testing memory consistency in WebGPU Shader Language (WGSL) [43, 44]. FusionRender [10] enhances end-to-end performance by merging object signatures in WebGPU. NNJit [35] enables just-in-time (JIT) auto-generation of optimized

WGSL kernels for edge devices. Ferguson et al. explored cache attacks in WebGPU to identify browser clients through side-channel attacks [21]. Giner et al. employed side-channel attack techniques to conduct memory leakage attacks on WebGPU [25]. Practical applications of WebGPU are also studied [12, 17, 32, 33, 56, 67]. GL2GPU focuses on dynamically translating WebGL applications to WebGPU within the JavaScript runtime to boost end-to-end rendering performance.

**Graphics API Mapping Frameworks.** Many efforts have been made to implement API mapping at the OS level. ANGLE [26] (Almost Native Graphics Layer Engine) translates OpenGL ES [40] calls to other graphics API backends like Direct3D 9 or OpenGL. Recently, ANGLE has expanded its support to include modern graphics APIs such as Vulkan [38] and Metal [9]. As a driver for OpenGL ES, ANGLE requires adaptation for each specific graphics rendering backend API, necessitating substantial development and maintenance effort. Other API transition layers that preserve the programming model include Vkd3d [30] and MoltenVK [39]. MoltenVK maps Vulkan API calls to their Metal equivalents, allowing applications originally designed for Vulkan to run on Apple's Metal framework without altering the underlying programming paradigm. Similarly, Vkd3d maps Direct3D 12 calls to Vulkan, enabling applications designed for Direct3D 12 to leverage Vulkan's capabilities without requiring changes to the original Direct3D 12 programming model. In contrast, GL2GPU translates WebGL applications to WebGPU within the JavaScript runtime, eliminating the need for developing additional drivers for specific devices to harness the advantages of WebGPU.

**JavaScript Prototype Patching.** Researchers use JavaScript's inherent flexibility to implement language-level (i.e., in-band) instrumentation within JavaScript applications during runtime. Roesner et al. [58] devised a client-side approach for detecting and categorizing common web tracking techniques. Besides, JavaScript Zero [60] leverages sophisticated JavaScript functionalities to identify potential attacks. Furthermore, ObjLupAnsys [46] detects prototype pollution vulnerabilities through object lookup analysis. Additionally, OpenWPM [18] utilizes instrumentation to gather data across various websites. Unlike previous research, our approach utilizes JavaScript prototype patching to capture WebGL invocations and translate them into WebGPU.

## 6 Conclusion

In this paper, we have proposed GL2GPU, a novel dynamic WebGL to WebGPU translator of web graphics rendering applications. We have designed mechanisms to maintain rendering consistency and enhance rendering performance significantly. Experimental results have demonstrated that GL2GPU preserves visual consistency and significantly improves rendering performance. Our ablation study has validated the effectiveness of the performance optimization mechanisms. The extension of GL2GPU to further application-level optimizations (e.g., bundle prediction) constitutes an exciting avenue for future works. By bridging the gap between the widespread WebGL and the advanced performance of WebGPU, GL2GPU aims to enhance the current web ecosystem in next-generation multimedia and inspire future research in this rapidly evolving field.

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

## A  Web Graphics Rendering API

### A.1  WebGL

Introduced by the Khronos Group in 2011 [29], WebGL is a high-performance graphics API for web browsers, derived from OpenGL ES (OpenGL for Embedded Systems) [28]. WebGL manages the rendering context using a global state[1], which encapsulates all configurations necessary for rendering. Developers establish the rendering context in WebGL by initiating a global state and defining essential rendering parameters such as color, depth testing,

---

[1] A detailed visualization diagram of this global state can be viewed at https://webglfundamentals.org/webgl/lessons/resources/webgl-state-diagram.html.

and blending modes. They activate shaders to process scene geometry, create vertex buffers for defining object shapes, manage texture mapping, and set uniforms and attributes to transfer data to the shaders, ensuring objects are rendered with accurate visual properties.

Once the necessary states are configured, the rendering process can begin. The WebGL driver leverages these states to execute drawing commands, directing the GPU to render the defined geometry using the current state configuration, shaders, and textures. This process culminates in the generation of pixels on the canvas, resulting in the final image being displayed.

### A.2  WebGPU

Introduced in 2017 [68], WebGPU is a cutting-edge web-based graphics and computation API that leverages modern GPU capabilities. The API design of WebGPU draws inspiration from modern APIs such as Vulkan[38], Direct3D 12[51], and Metal[9]. Unlike WebGL, which uses a global state to configure rendering settings, WebGPU organizes rendering configurations into more independent resources. This granular management of resources enhances the reusability of WebGPU and improves rendering performance.

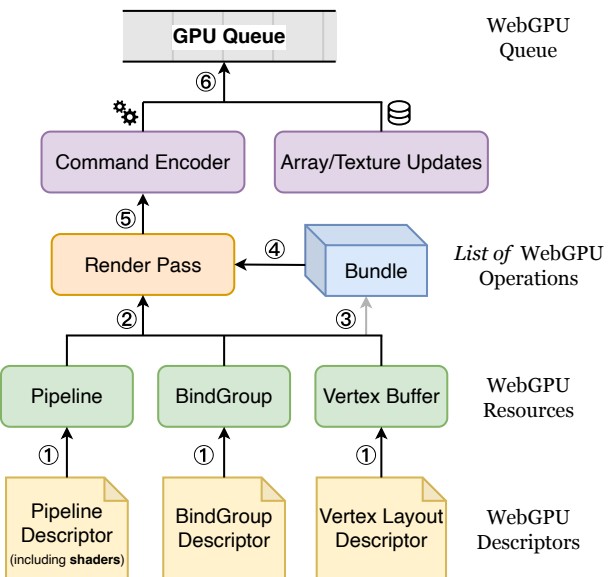

**Figure 6: The rendering workflow of WebGPU.**

As shown in Figure 6, resource creation in WebGPU begins with defining properties in descriptors (①), including `Pipeline`, `BindGroup`, and `VertexBuffer`. A `Pipeline` controls the GPU's vertex and fragment shader stages, bind groups define data usage in shader stages, and vertex buffers store graphical vertex attributes. The `BindGroup` defines how the data are used in shader stages, and the vertex buffer contains the graphical vertex attributes. Next, resources are packed into a `RenderPass` (②), which describes a sequence of GPU-executed rendering operations. Additionally, resources can also be packed into a bundle (③), which is then loaded into the RenderPass (④), and passed to the command encoder (⑤). The command encoder encodes commands for GPU execution, and

the resulting commands, along with necessary buffers and textures, are submitted to the GPU (⑥). The GPURenderBundle [49] in WebGPU is a partially encoded RenderPass that can be executed multiple times within future RenderPasses. This feature, coupled with JavaScript operations like setBindGroup() and setPipeline(), reduces CPU overhead by allowing the reuse of pre-encoded commands, optimizing resource utilization and enhancing rendering efficiency.

A bundle (named GPURenderBundle[49] in WebGPU API) is a partially RenderPass that is encoded once and can subsequently be executed multiple times within future RenderPass-es. The process of packing resources into a RenderPass is achieved through WebGPU operations, which is a series of JavaScript calls, such as setBindGroup() and setPipeline(). The introduction of the bundles aims to reduce the CPU overhead associated with packing WebGPU operations within the JavaScript runtime. This unique feature optimizes overall resource utilization and enhances rendering efficiency by enabling developers to reuse pre-encoded commands effectively.

