# OpenReview forum: "GL2GPU: Accelerating WebGL Applications via Dynamic API Translation to WebGPU"
_ACM.org/TheWebConf/2025/Conference — WWW 2025 Oral_

### Official Review · Reviewer_sBEE · 2024-11-26

**Novelty:** 4
**Technical Quality:** 4

**Review:**

The paper introducing **GL2GPU**, an innovative intermediate layer that dynamically translates WebGL applications to WebGPU, showcases high-quality research with a clear exposition of both technical details and experimental results. The authors have thoroughly articulated the design and implementation of GL2GPU, outlining its workflow and the functionalities of its various modules. By utilizing techniques such as JavaScript prototype patching, state caching, and uniform batching, GL2GPU effectively translates WebGL commands into their WebGPU counterparts, thereby enhancing rendering performance.

In terms of originality and significance, GL2GPU represents a pioneering effort to dynamically translate WebGL applications to WebGPU within the JavaScript runtime, all without modifying the browser or the original logic of existing WebGL applications. This innovative approach provides substantial benefits for web developers, as it enables them to harness the performance advantages of WebGPU without needing to rewrite their existing WebGL code. The significance of this work is underscored by its potential to address the performance gap between WebGL and WebGPU, an essential advancement for web graphics rendering technology.

However, the GL2GPU framework, while groundbreaking and potentially transformative for the web graphics rendering landscape, is not without its challenges. One primary concern is the additional complexity it introduces to the rendering pipeline, which may complicate maintenance and debugging for developers. Moreover, the real-time API translation could lead to increased resource consumption, particularly in terms of CPU and memory usage, which might strain resource-constrained environments or complex graphics workloads. Additionally, compatibility issues have been identified, with the framework encountering difficulties in certain browsers. This indicates that broader adoption may necessitate extensive cross-platform testing and potential adaptations.

**Questions:**

- The paper details GL2GPU's innovative approach to dynamically translating WebGL to WebGPU through JavaScript prototype patching. Could you elaborate on how this mechanism effectively manages complex state changes in WebGL? Additionally, what challenges arise in ensuring that all state changes are accurately captured and translated?
- You have highlighted several impressive performance optimization strategies in GL2GPU, including state caching and render bundle management. My question pertains to the interaction between these optimizations: do they synergize effectively, or have you encountered instances where one optimization may diminish the benefits of another?
- The evaluation of GL2GPU across a diverse range of devices and browsers is commendable, demonstrating a concerted effort to ensure broad compatibility. Could you share any specific adjustments or workarounds implemented to address the varying levels of WebGPU support across these platforms?
- Your work illustrates GL2GPU's scalability with increasing object counts during benchmarks. However, I'm curious about its performance in applications that necessitate a high degree of interactivity or real-time graphics updates. How does GL2GPU manage such demanding scenarios?
- While GL2GPU provides notable improvements in runtime performance, I have concerns regarding the potential effects on application startup time due to the added translation layer. Could you discuss any measurements or observations you have made concerning the startup performance of applications utilizing GL2GPU?

**Reviewer Confidence:**

2: The reviewer is willing to defend the evaluation, but it is likely that the reviewer did not understand parts of the paper

**Scope:**

3: The work is somewhat relevant to the Web and to the track, and is of narrow interest to a sub-community

---

### Official Review · Reviewer_rSwk · 2024-12-02

**Novelty:** 5
**Technical Quality:** 4

**Review:**

GL2GPU is an excellent engineering practice. Through GL2GPU, WebGL can be dynamically converted to WebGPU, and several optimizations are employed to accelerate the execution. Nevertheless, the innovation in optimization is not sufficient. The designs are simple and straightforward.

My main concerns are the design details and organization of Section 3 and the baseline selection of the experimental part:

The conversion issues (3.2 and 3.3) have been discussed by numerous developers [1]. The design of 3.2 is quite straightforward. The work of translating GLSL into WGSL (3.3) has already been realized [2]. It would be better for the authors to compare their work with the existing solution.

The cache design (3.4) is plain and straightforward. The author ought to further elaborate on how the cache is designed in accordance with the characteristics of the conversion process, like evicting and prefetching, and discuss when the cache performs well and when the improvement is restricted.

Compared with the other designs, the authors allocate more space to Bundle management. Nevertheless, Bundle management offers limited performance improvement, and for some workloads, there is little or no improvement (Aquarium). The author should consider why this design is necessary or under what scenarios the design will bring about a significant performance enhancement.

The authors did not test the performance of WebGPU and the host resource consumption resulting from the use of GL2GPU. The authors emphasize that no code changes are necessary for WebGL applications. However, if the performance of GL2GPU is significantly worse than that of directly using WebGPU, the practicality of GL2GPU will be greatly diminished. If the performance of WebGPU is much superior to GL2GPU, then why not convert the WebGL project to a WebGPU project to obtain more performance benefits?

[1] https://hackernoon.com/migrating-from-webgl-to-webgpu
[2] https://github.com/gfx-rs/naga/

**Questions:**

The conversion issues (3.2 and 3.3) have been discussed by numerous developers. The design of 3.2 is quite straightforward. The work of translating GLSL into WGSL (3.3) has already been realized. It would be better for the authors to compare their work with the existing solution.

The cache design (3.4) is plain and straightforward. The author ought to further elaborate on how the cache is designed in accordance with the characteristics of the conversion process, like evicting and prefetching, and discuss when the cache performs well and when the improvement is restricted.

Compared with the other designs, the authors allocate more space to Bundle management. Nevertheless, Bundle management offers limited performance improvement, and for some workloads, there is little or no improvement (Aquarium). The author should consider why this design is necessary or under what scenarios the design will bring about a significant performance enhancement.

The authors did not test the performance of WebGPU and the host resource consumption resulting from the use of GL2GPU. The authors emphasize that no code changes are necessary for WebGL applications. However, if the performance of GL2GPU is significantly worse than that of directly using WebGPU, the practicality of GL2GPU will be greatly diminished. If the performance of WebGPU is much superior to GL2GPU, then why not convert the WebGL project to a WebGPU project to obtain more performance benefits?

**Reviewer Confidence:**

3: The reviewer is confident but not certain that the evaluation is correct

**Scope:**

4: The work is relevant to the Web and to the track, and is of broad interest to the community

---

### Official Review · Reviewer_GJ2z · 2024-12-02

**Novelty:** 5
**Technical Quality:** 6

**Review:**

### Paper Summary
This paper presents GL2GPU, an intermediate layer designed to dynamically translate WebGL API calls to WebGPU at JavaScript runtime, to improve the rendering performance of WebGL applications. The authors adopt three designs to address the inconsistency between WebGL and WebGPU programming models and the high translation overhead at JS runtime: (1) using JS prototype patching to track WebGL state changes, (2) caching WebGL states to reduce traversing overhead, and (3) reducing the overhead of recording rendering commands by using bundles.

### Strengthes
+ Easy to follow
+ Solid implementation, open-sourced code, and great demo
+ The evaluation results look promising

### Weaknesses
- The shaders are pre-translated from GLSL to WGSL, making GL2GPU not flexible to deal with new GLSL shaders.
- Some evaluations are missing.

**Questions:**

+ Could you provide the evaluation results of the overhead of the translation itself?

+ Is it possible to compare GL2GPU with some native baselines developed from existing work?

+ How are Layer-1 and Layer-2 caches maintained, e.g., cache size, cache eviction policy, etc.?

**Reviewer Confidence:**

2: The reviewer is willing to defend the evaluation, but it is likely that the reviewer did not understand parts of the paper

**Scope:**

4: The work is relevant to the Web and to the track, and is of broad interest to the community

---

### Official Review · Reviewer_zrdM · 2024-12-02

**Novelty:** 5
**Technical Quality:** 6

**Review:**

### Quality
The research has a high quality. The paper has a straight-forward claim: providing a translation layer to dynamically convert WebGL to WebGPU, leading to a higher rendering performance. The authors show that their approach works, produces a significant performance improvement and provide further details about the underlying structure of WebGPU which is interesting to read.

### Clarity
The paper is well written and easy to read. The authors have presented their ideas in a logical and organized manner.

### Originality
While the authors have explored a new topic (WebGPU), the idea to translate (or emulate) older interfaces to support new interfaces or hardware is not new. However, they have provided a comprehensive overview of WebGPU and its benefits, which adds value to the existing literature, as well as a working implementation.

### Significance
The significance of this paper is high. The adoption of WebGPU by web developers will lead to improved graphics performance, faster rendering times, and enhanced overall user experience. This study provides essential information for professionals
working with web-based graphics, as well as a practical implementation that can help to future proof existing applications.

The main pros of this paper are:
* Improved understanding of the rendering workflow in WebGPU
* Practical examples of dynamic API translation from WebGL to WebGPU
* Working implementation
* Comprehensive evaluation

**Questions:**

The paper does not list limitations. Are there any scenarios where an application of your framework does not work or is not beneficial?

**Reviewer Confidence:**

3: The reviewer is confident but not certain that the evaluation is correct

**Scope:**

4: The work is relevant to the Web and to the track, and is of broad interest to the community

---

### Official Review · Reviewer_HctL · 2024-12-02

**Novelty:** 7
**Technical Quality:** 7

**Review:**

The paper presents **GL2GPU**, an intermediate layer for dynamically translating WebGL to WebGPU at runtime. It addresses a critical need to bridge the performance gap between WebGL and the more advanced WebGPU API. The approach is well-motivated, and the authors provide a clear justification for focusing on runtime translation rather than static migration, which is labor-intensive. The evaluation of GL2GPU demonstrates impressive performance gains, making it a significant contribution to the field.

### **Strengths:**
1. Tackles a relevant and timely problem in the web graphics domain.
2. Provides measurable performance improvements (**45.05% reduction in average frame time**).
3. Ensures cross-platform applicability across devices and operating systems.
4. Maintains visual consistency while optimizing rendering performance.

### **Weaknesses and Areas for Improvement:**
1. A more detailed discussion on the overhead introduced by dynamic translation would strengthen the paper.
2. Including a comparison with native WebGPU implementations or other translation techniques could provide additional context and validation.

**Questions:**

1. How does **GL2GPU** handle WebGL features or extensions not supported by WebGPU?  Are there any limitations in the coverage?
2. What is the runtime overhead introduced by the dynamic translation layer, and how does it impact performance in complex applications?
3. How does GL2GPU compare with manually migrated WebGL applications using native WebGPU implementations?
4. Have you tested GL2GPU with real-world applications beyond the benchmarks used in the evaluation? If so, what were the results?
5. Can GL2GPU's approach be generalized for other APIs beyond WebGL and WebGPU?

**Reviewer Confidence:**

4: The reviewer is certain that the evaluation is correct and very familiar with the relevant literature

**Scope:**

4: The work is relevant to the Web and to the track, and is of broad interest to the community